# Assessing the Impact of SARS-CoV-2 Lineages and Mutations on Patient Survival

**DOI:** 10.3390/v14091893

**Published:** 2022-08-27

**Authors:** Carlos Loucera, Javier Perez-Florido, Carlos S. Casimiro-Soriguer, Francisco M. Ortuño, Rosario Carmona, Gerrit Bostelmann, L. Javier Martínez-González, Dolores Muñoyerro-Muñiz, Román Villegas, Jesus Rodriguez-Baño, Manuel Romero-Gomez, Nicola Lorusso, Javier Garcia-León, Jose M. Navarro-Marí, Pedro Camacho-Martinez, Laura Merino-Diaz, Adolfo de Salazar, Laura Viñuela, Jose A. Lepe, Federico Garcia, Joaquin Dopazo

**Affiliations:** 1Bioinformatics Area, Andalusian Public Foundation Progress and Health-FPS, 41013 Sevilla, Spain; 2Institute of Biomedicine of Seville, IBiS, University Hospital Virgen del Rocío/CSIC/University of Seville, 41013 Sevilla, Spain; 3Department of Computer Architecture and Computer Technology, University of Granada, 18011 Granada, Spain; 4GENYO, Centre for Genomics and Oncological Research, Pfizer/University of Granada/Andalusian Regional Government, PTS Granada, 18016 Granada, Spain; 5Subdirección Técnica Asesora de Gestión de la Información, Servicio Andaluz de Salud, 41001 Sevilla, Spain; 6Unidad Clínica de Enfermedades Infecciosas, Microbiología y Medicina Preventiva, Hospital Universitario Virgen Macarena, 41009 Sevilla, Spain; 7Departamento de Medicina, Universidad de Sevilla, C. San Fernando, 4, 41004 Sevilla, Spain; 8Centro de Investigación Biomédica en Red en Enfermedades Infecciosas (CIBERINFEC), ISCIII, 28029 Madrid, Spain; 9Servicio de Aparato Digestivo, Hospital Universitario Virgen del Rocío, 41013 Sevilla, Spain; 10Dirección General de Salud Pública, Consejería de Salud y Familias, Junta de Andalucía, 41020 Sevilla, Spain; 11Departamento de Metafísica y Corrientes Actuales de la Filosofía, Ética y Filosofía Política, Universidad de Sevilla, 41004 Sevilla, Spain; 12Servicio de Microbiología, Hospital Virgen de las Nieves, 18014 Granada, Spain; 13Instituto de Investigación Biosanitaria, ibs.GRANADA, 18012 Granada, Spain; 14Servicio de Microbiología, Unidad Clínica Enfermedades Infecciosas, Microbiología y Medicina Preventiva, Hospital Universitario Virgen del Rocío, 41013 Sevilla, Spain; 15Servicio de Microbiología, Hospital Universitario San Cecilio, 18016 Granada, Spain; 16FPS/ELIXIR-ES, Andalusian Public Foundation Progress and Health-FPS, 41013 Sevilla, Spain

**Keywords:** SARS-CoV-2, COVID-19, survival, virus genome, phylogeny

## Abstract

Objectives: More than two years into the COVID-19 pandemic, SARS-CoV-2 still remains a global public health problem. Successive waves of infection have produced new SARS-CoV-2 variants with new mutations for which the impact on COVID-19 severity and patient survival is uncertain. Methods: A total of 764 SARS-CoV-2 genomes, sequenced from COVID-19 patients, hospitalized from 19th February 2020 to 30 April 2021, along with their clinical data, were used for survival analysis. Results: A significant association of B.1.1.7, the alpha lineage, with patient mortality (log hazard ratio (LHR) = 0.51, C.I. = [0.14,0.88]) was found upon adjustment by all the covariates known to affect COVID-19 prognosis. Moreover, survival analysis of mutations in the SARS-CoV-2 genome revealed 27 of them were significantly associated with higher mortality of patients. Most of these mutations were located in the genes coding for the S, ORF8, and N proteins. Conclusions: This study illustrates how a combination of genomic and clinical data can provide solid evidence for the impact of viral lineage on patient survival.

## 1. Introduction

With more than 12 million sequences submitted to GISAID [1] and other databases, SARS-CoV-2 is probably one of the most widely sequenced pathogens. Successive waves of infection have resulted in the constant selection of SARS-CoV-2 variants with new mutations in their viral genomes [2,3,4]. Sometimes, these novel variants carry specific mutations that have been linked to higher transmissibility [5,6,7] and/or immune evasion [8,9], making them relevant from a public health perspective [10] and leading to their classification as variants of interest (VOI) or variants of concern (VOC) [11]. However, current studies have failed to provide solid evidence on the potential effects of viral variants or mutations on COVID-19 severity or patient survival. Paradoxically, the impact of host genetics over COVID-19 progression and patient survival [12], as recently revealed in case–control studies [13], genome-wide association studies [14,15,16,17], and whole-genome sequencing studies [18], is better known than the impact of the viral variants or the mutations present in the viral genome. For example, while some studies suggest that lineages as B.1.1.7 (alpha) are associated with increased mortality [19], others could not find such association [20,21]. Epidemiological studies suggest that certain mutations, such as the D614G mutation in the S protein, could be associated with higher mortality [22]. More recently, the delta variant was described as more transmissible and pathogenic than the alpha variant [23] and the omicron variant has been found to be more transmissible although less pathogenic than the delta variant [24,25]. Studies using undetailed patient outcomes (with no covariates considered) find some mutations potentially associated with severe COVID-19 [26]. Previously, a 382-nucleotide deletion in the open reading frame 8 was associated with milder infection [27]. Actually, the definition or variants of concern (VOC) or variants of interest (VOI) is proposed by the World Health Organization (WHO) [11], the Centers for Disease Control and Prevention (CDC) [28], and COVID-19 Genomics UK Consortium (COG-UK) [29] are based on observed transmissibility, greater severity of disease, or in vitro evidence of reduced antibody neutralization [30]. The phenotypes of these VOCs and VOIs depend on the presence of specific mutations, known as mutations of concern [31], found to be associated with higher transmissibility [5,6] and/or immune evasion [8,32]. However, because of the lack of large datasets in which viral genomes and detailed patient clinical data are simultaneously available, studies providing solid evidence on the effects of viral variants or mutations on COVID-19 severity or patient survival are scarce. Thus, there is an urgent need for the use of large clinical data repositories in combination with systematic viral genome sequencing to determine these relationships of high clinical relevance.

Andalusia, located in the south of Spain, is the third largest region in Europe; it has a population of 8.4 million, similar to a medium-sized European country such as Austria or Switzerland. In the beginning of the pandemic, Andalusia implemented an early pilot project for first-wave SARS-CoV-2 sequencing [33], which was later transformed into the genomic surveillance circuit of Andalusia [34,35], a systematic genomic surveillance program in coordination with the Spanish Health Authority. In addition, the Andalusian Public Health System has systematically been storing the EHRs data of all Andalusian patients in the Population Health Base (BPS, acronym from its Spanish name “base poblacional de salud”) [36] since 2001, making of this database one of the largest repositories of highly detailed clinical data in the world (containing over 13 million comprehensive registries) [36]. Data generated in both sequencing initiatives along with clinical data stored in BPS were used to carry out this study.

## 2. Materials and Methods

### 2.1. Design and Patient Selection

Among the whole-genome SARS-CoV-2 sequences obtained from the pilot project of SARS-CoV-2 sequencing [33] (in which 1000 viral genomes corresponding to the first wave, randomly sampled, representative of all the COVID-19 diagnosis in Andalusia between 19 February and 30 June 2020, were sequenced), the Spanish Genomic epidemiology of SARS-CoV-2 (SeqCOVID) [37], and the Genomic surveillance circuit of Andalusia [34,35] (including 2438 SARS-CoV-2 genomes corresponding to the second wave, systematically sequenced among PCR-positive individuals, following the recommendations of the Spanish Ministry of Health [38]), a total of 764 sequences corresponded to individuals hospitalized between 19 February 2020 and 30 April 2021. In particular, 287 samples corresponded to the pilot project, 103 to the SeqCOVID project, and 374 to the sequencing circuit.

### 2.2. Sequencing SARS-CoV-2 Genome

SARS-CoV-2 RNA-positive samples were subjected to whole-genome sequencing at the sequencing facilities of the Genyo (Granada, Spain), Hospital San Cecilio (Granada, Spain), Hospital Virgen del Rocío (Sevilla Spain), IBIS (Sevilla, Spain), and CABIMER (Sevilla, Spain). 

RNA preparation and amplification were performed as described in the protocols published by the ARTIC network [39] using the V3 version of the ARTIC primer set from Integrated DNA Technologies (Coralville, IA, USA). In brief, correlative amplicons covering the SARS-CoV-2 genome were created after cDNA synthesis by using SuperScript IV Reverse Transcriptase (Thermo Fisher Scientific, Waltham, MA, USA), 1 µL of random hexamer primers, and 11 µL of RNA. Libraries were prepared according to the COVID-19 ARTIC protocol v3 and Illumina DNA Prep Kit (Illumina, San Diego, CA, USA). Library quality was confirmed using the Bioanalyzer 2100 system (Agilent Technologies, Santa Clara, CA, USA). The libraries were then quantified by Qubit DNA BR (ThermoFisher Scientific, Waltham, MA, USA), normalized, and pooled, and sequencing was performed using Illumina MiSeq v2 (300 cycles) and NextSeq 500/550 Mid Output v2.5 (300 Cycles) sequencing reagent kits.

### 2.3. Sequencing Data Processing

Sequencing data were analyzed using in-house scripts and the nf-core/viralrecon pipeline software [40]. Briefly, after read quality filtering, sequences for each sample were aligned to the SARS-CoV-2 isolate Wuhan-Hu-1 (GenBank accession: MN908947.3) [41] using bowtie2 algorithm [42], followed by primer sequence removal and duplicate read marking using iVar [43] and Picard [44] tools, respectively. Genomic variants were identified through iVar software, using a minimum allele frequency threshold of 0.25 for calling variants and a filtering step to keep variants with a minimum allele frequency threshold of 0.75. Using the set of high confidence variants and the MN908947.3 genome, a consensus genome per sample was finally built using iVar.

With the aim of having all the genomic variants in our dataset, the whole set of consensus genomes, regardless of missing data, has been used as input to the Nextclade software [45]. Consensus genome was aligned against the SARS-CoV-2 reference genome and aligned nucleotide sequences were compared with the reference nucleotide sequence, one nucleotide at a time. Mismatches between the query and reference sequences are reported differently, depending on their nature: nucleotide substitutions, nucleotide deletions, or nucleotide insertions. Lineage assignment to each consensus genome was generated by the Pangolin tool [46]. 

The SARS-CoV-2 whole genomes are available in the European Nucleotide Archive (ENA) database under the project identifiers PRJEB44396, PRJEB47798, and PRJEB43166 (see also Appendix A).

The evolutionary rate of the virus was obtained using the Augur application [47]. Augur functionality relies on the IQ-Tree software [48], which estimates the phylogenetic tree by maximum likelihood using a general time-reversible model with unequal rates and unequal base frequencies [49], from which the evolutionary rate is inferred.

### 2.4. Clinical Data Preprocessing

Clinical data for 764 hospitalized patients was requested from the BPS. The data were transferred from BPS to the Infrastructure for secure real-world data analysis (iRWD) [50] from the Foundation Progress and Health, Andalusian Public Health System. 

The main primary outcome was COVID-19 death (certified death events during hospitalization). Following previous similar studies, the first 30 days of hospital stay were considered for survival calculations [51]. The time variable in the models corresponds to the length (in days) of hospital stay. Stays that imply one or more changes of hospital units are combined in a single stay where the admission and discharge dates were set to the start of the first and the end of the last combined stay, respectively. Finally, in order to reduce possible confounding effects due to reinfection mechanisms we have opted to include only the first stay for each patient. The data used from BPS to properly account for covariates known to be related with COVID-19 survival are listed in Table 1. 

### 2.5. Statistical Analysis

The statistical analysis has been performed at two levels, at the level of lineages and at the level of mutations in the viral genome. In order to elucidate the association between each lineage/mutation and the survival outcome, the following steps have been used: (i) as a first step a covariate balance analysis to determine the viability for a causal analysis was applied [52]; (ii) for these lineages or mutations suitable for causal analysis hazard ratios were obtained using the closed form variance estimator for weighted propensity score estimators with survival outcome [53]; (iii) a causal bootstrapped hazard ratio is also obtained for the same lineages or mutations [54].

In detail, the first step involved the use of inverse probability weighting (IPW) for each mutation/lineage. IPW is based on propensity scores generated using the *WeightIt* R package (v 0.12) [55], where the exposed condition is, in the case of lineages, being infected by a virus of a specific lineage and, in the case of viral mutations, being infected by a virus harboring a specific mutation. To assess the viability of a causal analysis based on IPW, the proportion of covariates that could be effectively balanced using the standardized mean differences test as implemented in the *Cobalt* R package (v 4.3.1) [56] was checked using the 0.05 threshold [52]. As covariates, variables previously associated with COVID-19 mortality were used, such as: age, sex, pneumonia/flu vaccination status, chronic obstructive pulmonary disease, hypertension, obesity, diabetes, chronic pulmonary and digestive diseases, asthma, chronic heart diseases, and cancer [57] (see Table 1).

Covariate-adjusted log hazard ratios (LHR) were computed for each mutation/lineage of interest using the closed form estimator as implemented in the *hrIPW* R package (v 0.1.3) [53]. For each analysis an estimate of the LHR along with a 95% confidence interval and a *p* value of significance was provided. This methodology provides a robust estimation of the variability of the LHR under IPW-based tests [53].

A mutation or a lineage is considered eligible for a causal analysis if the closed form estimator converges and all the covariates can be properly balanced.

In addition, a bootstrapped estimation of the covariate-adjusted LHR has been computed, where the causal adjustment has been done using IPW as follows: (i) the weights are computed with a binomial linear model where the response is the presence/absence of a given variant and the regressors are the covariates using *WeightIt*; (ii) a Cox proportional hazards model as implemented in the R package *Survival* (v 3.2); (iii) a bootstrapped 95% confidence interval of the LHR coefficient was computed using adjusted bootstrap percentile (BCa) as implemented in the R *Boot* package (v 1.3).

The theoretical *p*-values [53] associated with the survival outcome have been adjusted using the FDR method [58].

### 2.6. Visualization of Lineage Prevalence over Time

A script based on the CoVariants application [59] was used to visualize the distribution of lineage relative prevalence over the time period studied. Data from neighboring European countries (France, UK and Portugal) and Spain were obtained from GISAID [60].

## 3. Results and Discussion

Here, we used viral genomes from the pilot project of SARS-CoV-2 sequencing [33], the Genomic surveillance circuit of Andalusia [35], and the Spanish SeqCOVID project [37]. Among the individuals for whom a SARS-CoV-2 whole-genome sequence was available, 764 had a hospitalization event during the studied period, which covered 19 February 2020 to 31 April 2021. According to PANGO lineage classification [61], a total of 18 SARS-CoV-2 lineages were identified among the 764 viral sequences used in this study (see Appendix A), 5 of them were eligible for causal analysis (see Methods): A, A.2, B.1, B.1.177, and B.1.1.7. Figure 1 shows the circulation of different lineages in Andalusia and Spain during the studied period, and Appendix A shows the circulation in neighboring European countries. Although the different lineages emerged and declined approximately at the same time, documenting a fast and effective inter-country transmission, there are quantitative differences in their proportions. For example, B.1.1.177 was far more prevalent in Spain and Andalusia than in the surrounding countries (Portugal, France, and the United Kingdom (UK), see Appendix A). However, the fast substitution of the alpha lineage (B.1.1.7) was similar in all countries.

Figure 2 shows the log hazard ratios obtained for the different lineages. Only one of them, the alpha variant (B.1.1.7), has rendered a significant impact on patient survival (log Hazard Ratio, LHR, of 0.51, with a confidence interval (CI) = [0.14,0.88]). These results are in agreement with recent observations reporting that this variant suppresses the innate immune responses more effectively than first-wave isolates [62]. Interestingly, the A lineage, now virtually extinct, seems to cause a lower mortality than other lineages, although the result does not reach significance (LHR = −1.80, C.I. = [−3.84,0.19]). However, the retrospective survival analysis of lineages reveals relevant information on many lineages already extinct, or with very low representation, which limits its practical clinical application.

Contrarily, the survival analysis of mutations provides interesting information, given that a large proportion of the studied mutations are still present in current lineages. Moreover, it throws light on regions of the proteins in which mutations could be related to higher mortality. In total, 594 nucleotide mutations were found with respect to the SARS-CoV-2 reference genome [41], 49 of which were eligible for formal causal analysis (see Methods). Figure 3 represents the LHR of the different mutations, plotted along the structure of the protein (see also Appendix A). Among them, a total of 27 mutations presented a significant (FDR-corrected) association with patient survival, two of which have not been confirmed by subsequent bootstrapping analysis. Eighteen of them affect known Pfam [63] motifs (Table 2), some of them are related to relevant viral features. For example, S:T716I affects the PF01601 motif (coronavirus spike glycoprotein S2), which drives membrane penetration and virus cell fusion and is involved in host specificity [64]; ORF8:Y73C, ORF8:R52I, and ORF8:Q27*, which affect the PF12093 (betacoronavirus NS8 protein) motif, allowing SARS-CoV-2 ORF8 to form unique large-scale assemblies that potentially mediate unique immune suppression and evasion activities [65,66]; or S:N501Y, which affects the PF09408 motif (betacoronavirus spike glycoprotein S1, receptor binding), which has been implicated in binding to host receptors [67]. However, some motifs disrupted by mutations are of unknown function, such as PF19211 or PF12379, corresponding to NSP2 and NSP3 proteins, respectively, which suggests that other relevant viral functionalities not yet characterized could be affected. Moreover, one of the significant mutations, ORF1ab:I2230T, does not affect any known motif, but it is significantly associated with patient higher mortality (see Figure 2 and Appendix A) by itself, given that it does not present correlations with other mutations (see Appendix A). It is worth noting that some of these mutations associated with higher mortality in hospitalized unvaccinated patients are present in the current omicron variant, such as ORF1ab:del3674-3676, S:del69-70 and S:del144 in BA.1, and S:N501Y and S:P681H in BA.1 and BA.2. Although there are no direct comparisons between omicron and the variants present in the first wave, and the immunity status of the population was completely different, the delta variant approximately doubled the hospitalization ratio compared with alpha [23], while omicron only showed reduced severity compared to delta [24,25]. These mutations could contribute to this still higher pathogenicity, although it is difficult to interpret the effect of individual mutations in the context of new mutations without new clinical and genomic data.

Interestingly, some mutations in the viral genome seem to display a positive association with patient survival. The most remarkable case is the mutation ORF1ab:A3523V, which was significant with the bootstrap test (see Appendix A), although failed to be significant with the covariate-adjusted LHR test, because of the relatively small sample size. This mutation affects the 3C-like proteinase nsp5, a protein from the peptidase C30 family (Prosite domain PS51442), involved in the control of the activity of the coronavirus replication complex by processing ORF1ab and ORF1a into 16 non-structural proteins [70]. Because of this role, it has been suggested as a potential drug target for coronaviruses [70] and more recently for SARS-CoV-2 specifically [71]. Therefore, it could be speculated that less efficient replication might be behind the lower mortality associated with this mutation.

The interest on mutations has focused mainly on non-synonymous changes, which produce a modification of the protein sequence that may have a potential influence on SARS-CoV-2 phenotypic properties. In contrast, much less attention has been paid to synonymous changes, which has a less clear relationship with viral phenotypes; there are currently no reports of synonymous mutations of concern [30]. Here, for the first time, we describe nine synonymous mutations (G4300T, C2710T, C14676T, C15279T, C913T, C6968T, C5986T, C15240T, and T16176C) in the ORF1ab with a significant association to higher mortality in hospitalized COVID-19 patients (Figure 3). However, some of them can simply be highly correlated with other coding mutations (e.g., C15279T is highly correlated with ORF1ab:T5303T, and C15240T is correlated with ORF1ab:T1567A), as depicted in Appendix A. Lineages harbor specific mutational profiles that are inherited by the descendants, along with some new mutations, thus creating a pattern of correlation between the mutations characteristic of lineages.

The evolutionary rate displayed by SARS-CoV-2 since February 2020 in the Andalusia region, according to the SARS-CoV-2 whole-genome sequencing circuit [34,35], is of 0.00063 substitutions per nucleotide per year (s/n/y), in agreement with the evolutionary rate previously described, which ranged from 0.0004 and 0.002 s/n/y [2,4,30,72,73]. Interestingly, when mutations associated with high mortality (such as ORF1ab: A1708D) are depicted over the clock-adjusted phylogeny, these tend to appear in the variants that have shaped the evolution of the virus in the Andalusia region during the period under study, with many of them related to the alpha (B.1.1.7) lineage (See Appendix A). The mutation associated with the highest mortality (ORF1ab:T1567A) shows a similar evolutionary rate (see Appendix A) and it seems to define a specific clade within the alpha lineage (Appendix A). However, some specific mutations, such as those marginally associated with better survival (e.g., N:D377Y), appear in variants with apparently slower mutation rates (B.1.177, and sublineages), although it also appears in lineages B.1 and A,2, which are now extinct, and in a few variants that are ancestors of the delta lineage. Actually, all the sublineages of the delta lineage carry this mutation, according to the Genomic surveillance circuit of Andalusia [35] (see [74] and Appendix A).

## 4. Conclusions

To summarize, the combined use of SARS-CoV-2 genome sequences and detailed clinical information of the patients allowed us to assess the impact of both the SARS-CoV-2 lineage and the mutations each virus harbors on the mortality rate among patients hospitalized for COVID-19. These studies provide a more realistic and unbiased approach to define VOIs and VOCs.

## Figures and Tables

**Figure 1 viruses-14-01893-f001:**
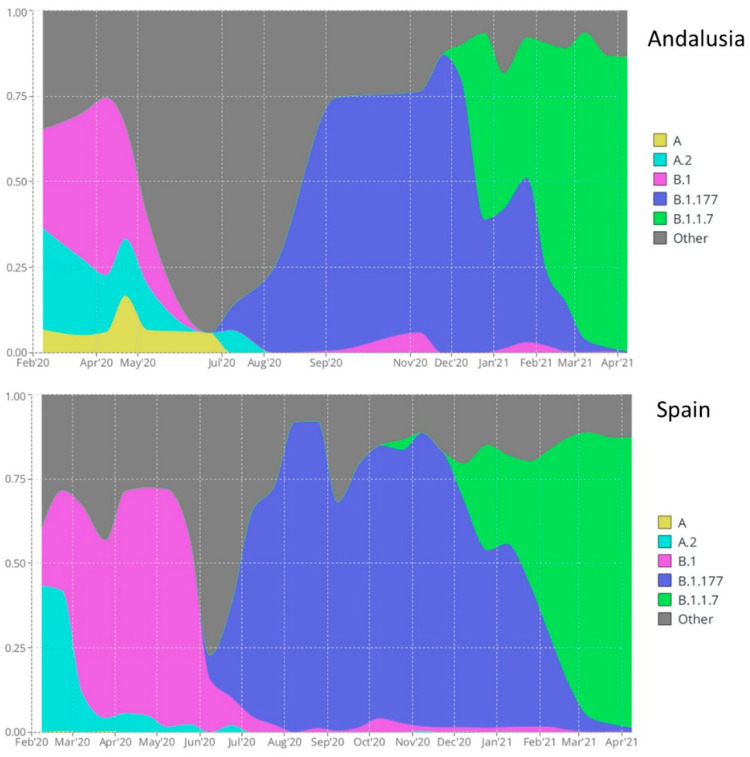
Circulation of the five SARS-CoV-2 variants eligible for the causal analysis in Andalusia (**upper panel**) and Spain (**lower panel**).

**Figure 2 viruses-14-01893-f002:**
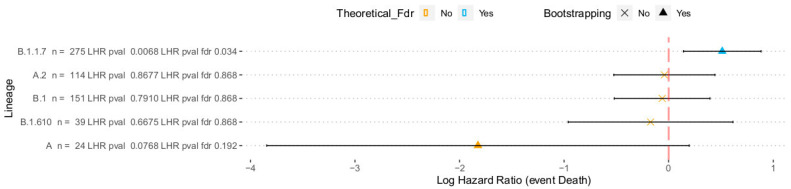
Log hazard ratios estimated for the five variants eligible for the causal analysis using the two approaches described in the text (the closed form estimator and the bootstrap). For each analysis, an estimate of the LHR along with a 95% confidence interval and a *p*-value (FDR-adjusted) of significance is provided.

**Figure 3 viruses-14-01893-f003:**
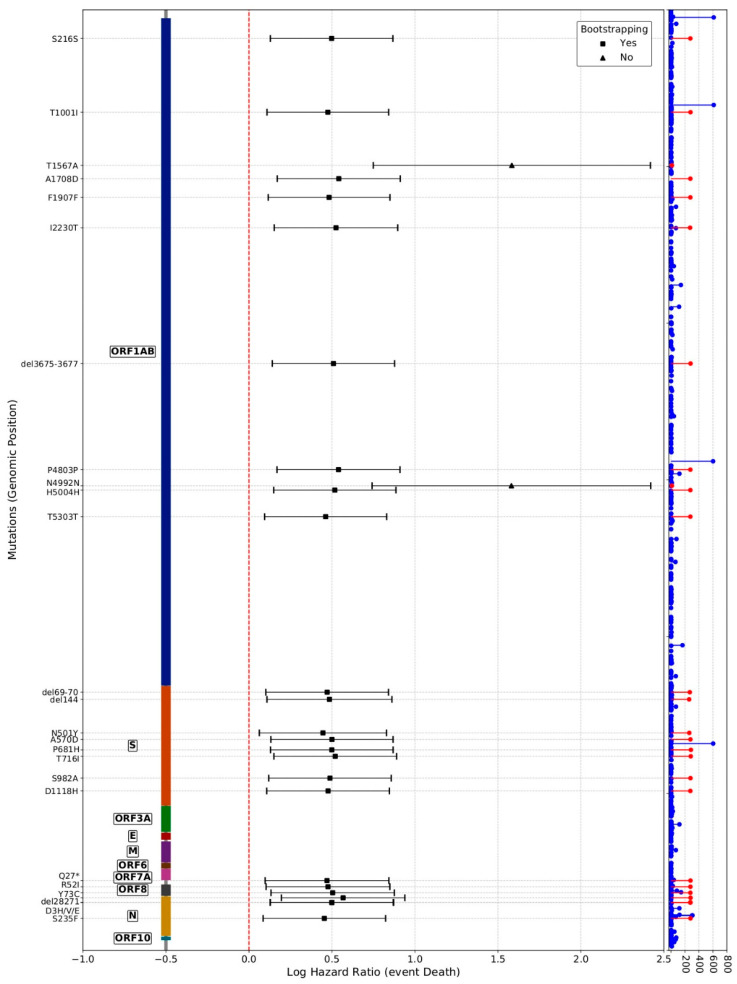
Log hazard ratios estimated for the 25 viral mutations that presented a significant (FDR-corrected) association with patient survival. Causal analysis was carried out using the two approaches described in the text (the closed form estimator and the bootstrap). For each analysis, an estimate of the LHR along with a 95% confidence interval and a *p*-value (FDR-adjusted) of significance is provided. Mutations are represented over the genomic positions in which they occur and on the left part, the corresponding proteins are annotated. The right part represents the observed distribution of mutations observed in all the samples analyzed.

**Table 1 viruses-14-01893-t001:** Data imported from BPS for each patient: code and definition of the variable.

Code	Meaning
FECNAC	Birth date
FECDEF	Death date
SEXO	Gender
FEC_INGRESO	Hospital admission date
FEC_ALTA	Discharge date
MOTIVO_ALTA	Reason for the discharge: (recovery/death/admission in another hospital/voluntary discharge/retirement home/unspecified)
COD_PATOLOGIA_CRONICA	Hospital codes for chronic conditions
COD_FEC_INI_PATOLOGIA	Date of condition diagnosis
COD_CIE_NORMALIZADO	A mixture of ICD9 and ICD10 codes for diseases
DESC_CIE_NORMALIZADO	Description of the ICD
FECINI_DIAG	Diagnosis date
FECFIN_DIAG	End of the diagnosed condition
FUENTE_DIAG	Source of the diagnosis (hospital, emergency, etc.)
IND_CRONICO_HCUP	Is it a chronic disease? (yes/no)
Test COVID: FECHA	Test COVID date
Test COVID: TYPE	PCR/antigens
Test COVID: RESULTADO_TEST	Result of the test (positive/negative)
Pharmacy (Hospital and external): DESCRIPCION	List of drugs used in hospital or purchased in the pharmacies
Pharmacy (Hospital and external): FECHA	Dispensing date
VACUNA	List of vaccines
VACUNAFECHA	Vaccination dates

**Table 2 viruses-14-01893-t002:** Mutations associated with higher patient mortality that affect PFAM motifs and pangolin lineages eligible for causal analysis with the mutation (non-synonymous from outbreak.info [68] and synonymous from cov-spectrum.org [69].

Mutation	Position	CDS	AAc Position	AAc Mutation	PFAM ^1^	Definition	Lineages Eligible for Causal Analysis Bearing the Mutation
C3267T	3267	ORF1ab	1001	ORF1ab:T1001I	PF12379	Betacoronavirus replicase NSP3, N-terminal	A; B.1.177; B.1.1.7
A4964G	4964	ORF1ab	1567	ORF1ab:T1567A	PF08715	Coronavirus papain-like peptidase	B.1; B.1.1.7
C5388A	5388	ORF1ab	1708	ORF1ab: A1708D	PF08715	Coronavirus papain-like peptidase	B.1; B.1.177; B.1.1.7
del11288. 11297	11288	ORF1ab	3975-3677	ORF1ab:del3675-3677	PF08717	Coronavirus replicase NSP8	A; A.1; B.1; B.1.177; B.1.1.7
C14676T	14676	ORF1ab	4803	ORF1ab:P4803P	PF00680	RNA-dependent RNA polymerase	B.1; B.1.177; B.1.1.7
C15279T	15279	ORF1ab	5004	ORF1ab: H5004H	PF00680	RNA-dependent RNA polymerase	B.1; B.1.177; B.1.1.7
del21766.21772	21766	S	69-70	S:del69-70	PF16451	Betacoronavirus-like spike glycoprotein S1, N-terminal	A; A.1; B.1; B.1.177; B.1.1.7
del21994.21997	21994	S	144	S:Y144-	PF16451	Betacoronavirus-like spike glycoprotein S1, N-terminal	A; A.1; B.1; B.1.177; B.1.1.7
A23063T	23063	S	501	S:N501Y	PF09408	Betacoronavirus spike glycoprotein S1, receptor binding	A; A.1; B.1; B.1.177; B.1.1.7
C23271A	23271	S	570	S:A570D	PF19209	Coronavirus spike glycoprotein S1, C-terminal	A; B.1; B.1.177; B.1.1.7
C23709T	23709	S	716	S:T716I	PF01601	Coronavirus spike glycoprotein S2	B.1; B.1.177; B.1.1.7
T24506G	24506	S	982	S:S982A	PF01601	Coronavirus spike glycoprotein S2	B.1; B.1.177; B.1.1.7
G24914C	24914	S	1118	S:D1118H	PF01601	Coronavirus spike glycoprotein S2	B.1; B.1.177; B.1.1.7
C27972T	27972	ORF8	27	ORF8:Q27*	PF12093	Betacoronavirus NS8 protein	A; B.1; B.1.177; B.1.1.7
G28048T	28048	ORF8	52	ORF8:R52I	PF12093	Betacoronavirus NS8 protein	A; B.1; B.1.177; B.1.1.7
A28111G	28111	ORF8	73	ORF8:Y73C	PF12093	Betacoronavirus NS8 protein	A; B.1; B.1.177; B.1.1.7
C28977T	28977	N	235	N:S235F	PF00937	Coronavirus nucleocapsid	A; B.1; B.1.177; B.1.1.7

^1^ PFAM information can be accessed at: https://pfam.xfam.org/family/PFXXXX with PFXXXX being the corresponding PFAM ID.

## Data Availability

The sequences of the SARS-CoV-2 whole genomes presented here are available in the European Nucleotide Archive (ENA) database under the project identifiers PRJEB44396, PRJEB47798, and PRJEB43166. Appendix A contains a detailed list of individual ENA IDs per sequence.

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
