# Peer review of "Assessing the Impact of SARS-CoV-2 Lineages and Mutations on Patient Survival"

_viruses, 2022, doi:10.3390/v14091893_

Round 1

Reviewer 1 Report

Minor comments

Line 44. S, ORF8 and N genes. Mutations occur in genes not proteins, correct throughout the manuscript.

Line 210 B.1 instead of B1.

Line 212 United Kingdom (UK) instead of UK.

Major comments

Line 123. Please include in supplementary information the GISAID EPI accession numbers with which this analysis is being performed, in order to someone wants to replicate it.

Lines 198 and 264. For a more robust analysis of the mutations and their association with hospitalization, I suggest a validation of your findings with samples from the same analysis period using GISAID, as there are cases uploaded to this platform that include the patient's status, e.g. hospitalized.

Line 255. How can this be explained? Were there more hospitalizations with the omicron BA.1 and BA.2 variant?

Line 265. Detail the implication of this mutation on the function of the protein, is there any papers on this?

Line 276. How could this correlation be explained?

Muy interesante trabajo, felicidades.

Author Response

We appreciate very much the constructive comments of the referee that have contributed to increase the quality of the manuscript. See below the point-by-point responses to them:

COMMENT

=========

Line 44. S, ORF8 and N genes. Mutations occur in genes not proteins, correct throughout the manuscript.

RESPONSE

=========

The referee is right. We have reformulated the sentence to “Most of these mutations were located in the genes coding for S, ORF8 and N proteins” and corrected throughout the text.

COMMENT

=========

Line 210 B.1 instead of B1.

RESPONSE

=========

Apologies for the oversight. Fixed.

COMMENT

=========

Line 212 United Kingdom (UK) instead of UK.

 RESPONSE

=========

Fixed

Major comments

COMMENT

=========

Line 123. Please include in supplementary information the GISAID EPI accession numbers with which this analysis is being performed, in order to someone wants to replicate it.

RESPONSE

=========

We have submitted the data to ENA. We have added the sentence “The SARS-CoV-2 whole genomes are available in the European Nucleotide Archive (ENA) database under the project identifiers PRJEB44396, PRJEB47798 and PRJEB43166 (see also Supplementary Table S1)” in current line 140, after the description of the whole sequencing data processing. In any case, there was a reference to the data in the “data availability” section, but we agree with the referee in the fact that making a reference in the “2.3 Sequencing data processing” section makes it clearer.

COMMENT

=========

Lines 198 and 264. For a more robust analysis of the mutations and their association with hospitalization, I suggest a validation of your findings with samples from the same analysis period using GISAID, as there are cases uploaded to this platform that include the patient's status, e.g. hospitalized.

RESPONSE

=========

We agree with the referee that this validation could be a very good addition to the manuscript. We have inspected the data available in GISAID corresponding to the period analyzed (19/02/2020 a 30/04/2021) and there are about 1.5 million sequences. A total of 76,760 of them have some information on the patients. However, the information stored is very scarce, low quality (no standard vocabulary has been used) and does not include relevant clinical information on variables that are known to be associated with death or serious complications (see Table 1 in the manuscript). Essentially there is information on gender and age and some extra information on hospitalization (11492) and death (2573). Since no standard vocabulary has been used, there are more than 30 ways in which these data have been introduced (e.g. death, deceased, demised, dead, etc. including many typos). The lack of information on other critical variables (see section 2.5) related to COVID19 mortality makes the covariate-adjusted survival analysis impossible and consequently the validation. Any correlation obtained between death and viral variants or mutations will be affected by unknown biases due to the covariates not considered, which will invalidate the results. Having access to detailed and high-quality clinical data is precisely the reason for which we used the Andalusian Population Health Database and what makes this study unique to understand the real link between viral genomic profiles and patient survival.

COMMENT

=========

Line 255. How can this be explained? Were there more hospitalizations with the omicron BA.1 and BA.2 variant?

RESPONSE

=========

This is a very good point. According to published reports, there was an increase of the hospitalizations in absolute terms because of the increase of the transmissibility, but the hospitalization ratio of omicron (BA.1 and BA.2) is lower than those observed for delta (OR 0.34 in unvaccinated patients) (Sievers et al, Eurosurveillance). However, the risk of hospitalization of the delta variant was more than double that the alpha (Elliot et al, Science). So, it might be that omicron was still more pathogenic than alpha. Unfortunately, we could not find direct comparisons between alpha and omicron. However, this is very difficult to say because the effect of the individual mutations is modulated by the context of the rest of mutations in the carrier virus. We have added a sentence in the manuscript explaining this.

COMMENT

=========

Line 265. Detail the implication of this mutation on the function of the protein, is there any papers on this?

RESPONSE

=========

We could not find any paper that commented on this specific mutation (beyond some tables with plain descriptions of its occurrence in some isolates). The mutation corresponds to the 3C-like proteinase nsp5, a protein from the Peptidase C30 family, involved in the control of the activity of the coronavirus replication complex, by processing ORF1ab and ORF1a into 16 non-structural proteins. This protein has been reported as a potential drug target (Dai et al., Science). We can only speculate that it could negatively affect the replicative potential of the virus, making it less infective. We have included a sentence in the text explaining this.

COMMENT

=========

Line 276. How could this correlation be explained?

RESPONSE

=========

Thanks for bringing this question. This is because the mutations occur together with other previous mutations in the virus, and all together are inherited by the descendance. However, it is worth explaining this in the text. We have included a sentence with this purpose.

COMMENT

=========

Muy interesante trabajo, felicidades.

RESPONSE

=========

Thanks a lot. We appreciate this comment very much.

Reviewer 2 Report

1) In the introduction the authors mention the variants of concern but do not mention at any point the current situation in 2022, a mention of omicron variants would be interesting

2) The references are dated 2021 for the most recent ones, in a field like COVID-19, the literature evolves rapidly. An update of the references is therefore necessary.

3) The authors refer to table 2 but no table 2 is available in the manuscript. 

4) All the figures (manuscript and supplemntal material) need to be revised and reworked in order to have a satisfactory quality which is not currently the case.

5) English must be improve

6) The sentence from line 111 to line 116 is not clear.

7) Has there been any follow-up to the study during 2022? If so, it would be interesting to integrate it into the manuscript to make it more current

Author Response

We appreciate very much the constructive comments of the referee that have contributed to increase the quality of the manuscript. See below the point-by-point responses to them:

COMMENT

=========

1) In the introduction the authors mention the variants of concern but do not mention at any point the current situation in 2022, a mention of omicron variants would be interesting

RESPONSE

=========

The referee is right. We have included a sentence mentioning the delta and the recent omicron variants.

COMMENT

=========

2) The references are dated 2021 for the most recent ones, in a field like COVID-19, the literature evolves rapidly. An update of the references is therefore necessary.

RESPONSE

=========

We have updated some of the references (7, 9, 17, 23, 24, 25) to add more recent context in the introduction. We appreciate the comment of the referee because that will increase the quality of the manuscript.

COMMENT

=========

3) The authors refer to table 2 but no table 2 is available in the manuscript.

RESPONSE

=========

Our apologies, it was as Table 3. We have fixed it.

COMMENT

=========

4) All the figures (manuscript and supplemntal material) need to be revised and reworked in order to have a satisfactory quality which is not currently the case.

RESPONSE

=========

The figures in the manuscript are embedded in the word file that has later been converted to PDF and they have been sort of compressed. The original figures in vectoral format and more quality have been uploaded.

COMMENT

=========

5) English must be improve

RESPONSE

=========

We have revised the English carefully trying to improve it. We hope now is more acceptable.

COMMENT

=========

6) The sentence from line 111 to line 116 is not clear.

RESPONSE

=========

We have rewritten the sentence. We think now is clearer.

COMMENT

=========

7) Has there been any follow-up to the study during 2022? If so, it would be interesting to integrate it into the manuscript to make it more current

RESPONSE

=========

Unfortunately, there is not a follow up yet. The ethics committee allowed us to collect data only from the dates of the current study. We are planning to extend the study in the future, but this is still in discussion.  

Round 2

Reviewer 1 Report

The authors made the indicated corrections. 

Reviewer 2 Report

I have no more comment